# Seroprevalence of West Nile Virus among Equids in Bulgaria in 2022 and Assessment of Some Risk Factors

**DOI:** 10.3390/vetsci11050209

**Published:** 2024-05-09

**Authors:** Nikolina Rusenova, Anton Rusenov, Mihail Chervenkov, Ivo Sirakov

**Affiliations:** 1Department of Veterinary Microbiology, Infectious and Parasitic Diseases, Faculty of Veterinary Medicine, Trakia University, 6000 Stara Zagora, Bulgaria; 2Department of Internal Diseases, Faculty of Veterinary Medicine, Trakia University, 6000 Stara Zagora, Bulgaria; anton.rusenov@trakia-uni.bg; 3Faculty of Veterinary Medicine, University of Forestry, 1797 Sofia, Bulgaria; vdmchervenkov@abv.bg; 4Institute of Biodiversity and Ecosystem Research, Bulgarian Academy of Sciences, 1113 Sofia, Bulgaria; 5Department of Medical Microbiology, Faculty of Medicine, Medical University-Sofia, 2 Zdrave Str., 1431 Sofia, Bulgaria; insirakov@medfac.mu-sofia.bg

**Keywords:** WNV, horses, donkeys, ELISA, VNT, Usutu virus, seroprevalence

## Abstract

**Simple Summary:**

This study aimed to analyze the seroprevalence of West Nile virus (WNV) among equids in Bulgaria; confirm the results of two tests, ELISA and the virus neutralization test (VNT); and investigate some predisposing factors for WNV seropositivity. A total of 378 serum samples from 15 provinces in northern and southern Bulgaria were tested. Fifteen samples were positive in both assays, i.e., the seroprevalence was 3.97%. When compared with VNT, ELISA showed 100.0% sensitivity and 94.5% specificity. The analysis of risk factors showed that the region, altitude of the locality, type of housing and breed were associated with WNV seropositivity. The results from this study demonstrate that WNV circulates among equids in Bulgaria and that they could be used to determine the respective risk levels in provinces or regions of the country.

**Abstract:**

The aim of this study was to analyze the seroprevalence of West Nile virus (WNV) among equids in Bulgaria, confirm the results of a competitive ELISA versus the virus neutralization test (VNT) and investigate some predisposing factors for WNV seropositivity. A total of 378 serum samples from 15 provinces in northern and southern Bulgaria were tested. The samples originated from 314 horses and 64 donkeys, 135 males and 243 females, aged from 1 to 30 years. IgG and IgM antibodies against WNV protein E were detected by ELISA. ELISA-positive samples were additionally tested via VNT for WNV and Usutu virus. Thirty-five samples were WNV-positive by ELISA (9.26% [CI = 6.45–12.88]), of which 15 were confirmed by VNT; hence, the seroprevalence was 3.97% (CI = 2.22–6.55). No virus-neutralizing antibodies to Usutu virus were detected among the 35 WNV-ELISA-positive equids in Bulgaria. When compared with VNT, ELISA showed 100.0% sensitivity and 94.5% specificity. A statistical analysis showed that the risk factors associated with WNV seropositivity were the region (*p* < 0.0001), altitude of the locality (*p* < 0.0001), type of housing (*p* < 0.0001) and breed (*p* = 0.0365). The results of the study demonstrate, albeit indirectly, that WNV circulates among equids in northern and southern Bulgaria, indicating that they could be suitable sentinel animals for predicting human cases and determining the risk in these areas or regions of the country.

## 1. Introduction

West Nile fever is a zoonosis that is caused by a single-stranded, positive-sense RNA virus from the *Orthoflavivirus* genus of the *Flaviviridae* family [1]. West Nile virus (WNV) belongs to the Japanese encephalitis serocomplex along with other antigenically related flaviviruses such as Murray valley encephalitis virus, Japanese encephalitis virus, Kunjin virus, Usutu virus, Kokobera virus, Alfuy virus and St. Louis encephalitis virus [2]. These viruses also belong to the arbovirus group, as they are transmitted by mosquitoes, mainly by *Culex* spp., and less frequently by other vectors [3]. Currently, there are nine genetic lineages of WNV that have been proposed based on biological and phylogenetic characteristics [4], and lineages 1 and 2 are the most virulent and widespread, including in Europe [5,6]. The transmission and maintenance cycle of WNV in nature has been well studied and documented in the scientific literature [7,8]. The main reservoirs of the virus are various wild bird species. Some of them develop the disease and die, while others become carriers without clinical manifestation. Mosquitoes transmit the virus to the susceptible population when sucking blood from a viremic host. Humans and equids are defined as “dead-end” hosts for WNV due to the low level of viremia, which is insufficient for further infection of vectors, resulting in the termination of the transmission cycle [9]. Humans and horses may develop a subclinical infection or clinical symptom complexes of varying severity, depending on the presence of concomitant factors. The available evidence suggests that approximately 1% of infected patients develop neuroinvasive disease including encephalitis and paralysis [10,11]. In horses, clinical signs vary, commonly including ataxia, weakness and muscle fasciculations [12,13].

The serological techniques that the World Organisation for Animal Health (WOAH) recommends for the detection of antibodies to WNV in different animal species are the plaque reduction neutralization test (PRNT), the micro-viral neutralization test (micro-VNT) and the ELISA immunoenzymatic assay [14]. Other assays, such as the complement fixation test and hemagglutination inhibition test, find less frequent use now, despite the excellent correlation found with the virus neutralization test [15]. WNV cases in horses usually precede those in humans [16]. Thus, serological surveillance in horses as sentinel animals is a useful epidemiological tool to predict human cases [17]. Data on the WNV seroprevalence in horses vary among studies in Europe, e.g., 7.1% in Andalusia (southern Spain) in 2010 [18], 0% to 13.77% in western and eastern regions of Germany in 2020 [19,20] and 8.6% in northern Serbia in 2013 [21].

There are a limited number of published reports on the circulation of WNV among equids in Bulgaria. Rusenova et al. [22] reported data on the WNV seroprevalence in donkeys and mules based on archival samples obtained in 2015. Other authors have reported virus circulation among humans in some regions of the country in 2015 and 2018 [23,24], among reservoirs (wild and migratory birds) [25] and among vectors (*Culex pipiens*) [26].

The aim of this study was to provide an update on the WNV seroprevalence among equids in Bulgaria and to explore the diagnostic potential of a commercially available competitive ELISA versus a virus neutralization test. In addition, some predisposing factors for WNV seropositivity were analyzed.

## 2. Materials and Methods

### 2.1. Samples

This study included a total of 378 serum samples collected in July–October 2022. They were from 314 horses and 64 donkeys, 135 males and 243 females, aged from 1 to 30 years (10.13 ± 6.58**/**mean ± SD). The samples were obtained from 25 municipalities in 15 provinces of Bulgaria: 19 in South Bulgaria and 6 in North Bulgaria. The provinces were located in the following six regions: Southeastern (n = 2), South Central (n = 3), Southwestern (n = 4), Northwestern (n = 2), North Central (n = 2) and Northeastern (n = 2). The tested animals were clinically healthy. Detailed descriptions of the tested samples by province, municipality, region, species and sex are presented in Appendix A. All samples were taken aseptically by jugular vein puncture (18-gauge needle) and collected in Vacutest collection tubes with clot activator (Vacutest Kima srl, Arzergrande, PD, Italy). After separation of the serum and centrifugation at 2000× *g* for 10 min, the samples were aliquoted into cryo tubes and stored at −80 °C until the testing in June 2023.

The owners of the animals or their keepers provided information about the age, sex, breed, type of housing (stalls, outdoor housing or outdoors with shelter) and vaccinations. Records were also kept of the altitude of the locality, the region and the month of sampling. Exclusion criteria: samples were not taken from animals that had moved from one province of the country to another within two years prior to the study.

### 2.2. Serological Study

The Ingezim West Nile Compac (Ingenasa, Madrid, Spain), a blocking enzyme immunoassay, was employed for testing samples, following the manufacturer’s instructions. The kit is designed to detect antibodies specific to the WNV glycoprotein E. Before use, the sera were heated at 56 °C for 45 min to inactivate thermolabile non-specific virus inhibitors. The optical density (OD) of each well was read at 450 nm on a Microplate Photometer (Biosan, Riga, Latvia) within 5 min after the addition of stop solution. The results were validated by calculating the average ODs of positive and negative controls, lower than 0.35 and higher than 0.8, respectively. Samples with an inhibition percentage (IP) of ≥40% were interpreted as positive and those with IP ≤ 30% were interpreted as negative, whereas samples with an IP between 30 and 40% were interpreted as doubtful.

Positive sera and a subset of negative sera (n = 5) were sent to the OIE Reference Laboratory for West Nile Disease in Teramo (Italy) for additional testing using a competitive IgG- and IgM-ELISA (ID Screen^®^ West Nile Competition Multi-species and ID Screen^®^ West Nile IgM Capture, IDvet, Grabels, France), following the company’s instructions. For validation in ID Screen^®^ West Nile Competition Multi-species, the criteria were as follows: mean value of the negative control OD (OD_NC_) > 0.700; mean value of the positive control OD_PC_/OD_NC_ < 0.3. Results interpretation: S/N % ≤ 40%—positive; 40% < S/N % ≤ 50%—doubtful; S/N % > 50%—negative. In ID Screen^®^ West Nile IgM Capture, the status of samples was interpreted as follows: S/P % ≤ 35%—negative; 35% < S/P % < 45%—doubtful; S/P % ≥ 45%—positive.

Positive and negative sera were also tested for the presence of IgG neutralizing antibodies against the West Nile virus and the closely related Usutu virus using the virus neutralization test at the reference laboratory. In summary, serial 2-fold dilutions starting from 1:5 were prepared in microtiter plates, and 100 tissue culture infective doses (TCID_100_) of antigen (strains Eg-101 and 939/01) were added to each dilution. Antigen/serum mixtures were incubated at 37 °C for 60 min, followed by the addition of 10^5^ Vero cells to each well. The plates were further incubated at 37 °C for 120 h, and starting from the 72nd hour, they were examined for cytopathic effects (CPEs) microscopically. The antibody titer was defined as the reciprocal of the highest dilution of the serum that showed 100% neutralization [27]. Positive and negative control sera were included in each plate. Samples with a neutralizing titer of 1:10 against WNV were considered positive. In line with common practice, in cases where neutralizing titers against both tested viruses were detected in a sample, the sample was interpreted as positive for the virus with a four-fold higher titer compared to the other tested virus. If a four-fold difference in neutralizing titers was not observed, the result was deemed inconclusive, and the samples were excluded from further analysis. Samples testing positive by competitive ELISA but negative by VNT were considered seronegative.

### 2.3. Statistical Analysis

The evaluation of the diagnostic performance of cELISA for WNV detection vs. the “gold standard” VNT was made through calculation of the area under the receiver operating characteristic (ROC) curve (AUC) as a measure of test diagnostic accuracy, sensitivity and specificity. The interpretation of the AUC as a measure of diagnostic accuracy was as follows: 0.90–1: excellent diagnostic test; 0.80–0.90: good diagnostic test; 0.70–0.80: fair diagnostic test; 0.60–0.70: poor diagnostic test; and 0.50–0.60: fail [28].

The differences between VNT-negative and VNT-positive animals for continuous variables (age, altitude) were tested by the Mann–Whitney non-parametrical test, whereas differences for categorical variables (species, breed, sex, type of housing, region, month) were tested by the *Chi*-square test. Non-parametric ROC curve analysis was performed using the recommended algorithm of DeLong et al. [29] to calculate the optimal cut-off value for dichotomization of the altitude on the basis of the Youden J statistic. Odds ratios (ORs) and 95% confidence intervals (CIs) for the ORs were calculated for each signalment parameter as a quantitative measure of its association with the outcome (WNV infection) by means of logistic regression analysis. All statistical analyses were performed using MedCalc 15.8 (Ostend, Belgium).

## 3. Results

Out of the 378 serum samples tested with the Ingezim West Nile Compac ELISA kit, 35 showed a positive result (9.26% [CI = 6.45–12.88]). The inhibition percentage of positive sera ranged from 76.20% to 99.95%. With the exception of one sample that was interpreted as doubtful, all others were confirmed by competitive IgG-ELISA in the reference laboratory. The initial c-ELISA testing at our laboratory did not yield doubtful results. Fifteen out of the thirty-five ELISA-positive samples were confirmed for WNV by the VNT in the reference laboratory; hence, the seroprevalence was estimated to be 3.97% (CI = 2.22–6.55) (Table 1). Neutralizing antibodies were detected in five of the studied provinces, with a prevalence ranging from 1.35% to 28.21%.

Figure 1 illustrates the sampling provinces and the presence of ELISA- and VNT-positive samples within them.

Neutralizing titers varied between 1:10 and 1:80. Nine animals showed a titer of 1:10 (60%), a titer of 1:20 was found in three animals (13.3%), a titer of 1:40 was found in three animals (20), and one animal had a titer of 1:80 (6.7%). No sera having specific antibodies against Usutu virus were detected (titer < 1:10). No samples containing neutralizing antibodies against both tested viruses were determined, nor were there any samples with inconclusive results; therefore, no sera were excluded from the subsequent analysis.

There was a non-significant relationship between VNT-WNV serological data and the animal’s age (*p* = 0.9125), species (horses/donkeys) (*p* = 0.9778) or sex (*p* = 0.6417). On the other hand, there was a significant association with the region (*p* < 0.0001), altitude (*p* < 0.0001), breed (*p* = 0.0365), type of housing (outdoor/shelter/indoor) (*p* < 0.0001) and month of sampling (*p* = 0.0150) (Table 2).

Figure 2 shows the diagnostic performance of cELISA as compared to the virus neutralization test. The area under the ROC curve (AUC) was 0.972, with a 95% confidence interval of 0.950 to 0.986 (*p* < 0.0001). The sensitivity and specificity of cELISA were 100.0% and 94.5%, respectively.

The results from the logistic regression analysis about the risk factors associated with WNV seroprevalence are presented in Table 3. The risk for animals raised in the Northwestern region of Bulgaria was 13 times higher (*p* = 0.0001) as compared to the Southwestern region. Animals in locations at an altitude below or equal to 109 m had a 16 times higher risk of infection compared to those at higher altitudes (*p* < 0.0001) (Figure 3). The risk for purebred animals was 5 times lower than that for mixed-breed animals (*p* = 0.0331). Equids that were provided shelter showed a 21 times higher risk of infection (*p* < 0.0005) than did those housed outdoors.

## 4. Discussion

The results from this study showed a WNV seroprevalence of 3.97% among equids in Bulgaria as detected by the virus neutralization test, the gold standard for the detection and differentiation of closely related flaviviruses [17,30,31]. Twenty cELISA-positive samples were not confirmed in the reference laboratory, which may be explained by the lower specificity of ELISA versus the virus neutralization reaction [32]. All ELISA-positive samples were further tested for neutralizing antibodies against Usutu virus, but no seropositivity was detected (titer < 1:10). Based on these results and based on previous studies [22,33], we can speculate that the circulation of Usutu virus among humans and other species in our country was not reported until 2022. The IP of the ELISA-positive samples showing virus-neutralizing antibodies was over 92%. However, we found samples with an IP of over 95.02% that were an exception, as no virus-neutralizing antibodies were detected against WNV or Usutu virus. According to the ELISA kit instructions, samples with IP ≥ 40% are interpreted as positive, and a virus neutralization test is recommended for doubtful results. These discrepancies motivated us to analyze the diagnostic capabilities of the ELISA compared with the gold standard for WNV antibody detection. ROC analysis showed an excellent diagnostic value of the cELISA used (AUC = 0.972), with 100% sensitivity and 94.5% specificity. A specificity of 95% was calculated by Beck et al. [17]; excellent sensitivity and specificity values of cELISA were also reported in [34]. Gómez-Vicente et al. [35] even found 100% specificity for IgG-ELISA when comparing its diagnostic capabilities against chemiluminiscent immunoassay in human serum samples. The differences between the two assays are probably due to cross-reactivity with other unstudied flaviviruses in Bulgaria or due to the high sensitivity of ELISA in the detection of antibodies against glycoprotein E in the early stages of infection, which are non-neutralizing. Although some positive results were unconfirmed by VNT, ELISA is a suitable method for the screening of WNV and other genetically similar flaviviruses due to its advantages such as its speed, its ability to test a large number of samples, no requirement for cell culture and no need of a biosafety level 3 laboratory to perform the assay [17].

There were IgG-ELISA-positive samples in 10 out of the 15 studied provinces in both southern and northern regions of Bulgaria, while VNT confirmed neutralizing antibodies in 5 out of these 10 provinces (Table 1, Figure 1). Neutralizing antibodies were not detected in the provinces of Stara Zagora, Burgas, Plovdiv, Smolyan and Gabrovo. As noted above, the sample with an IP of 95.02 originated from the Burgas region, which was a surprising finding due to the expected higher percentage of seroprevalence there compared to other regions tested. The province of Burgas is located in southeastern Bulgaria, bordered by the Black Sea to the east and Turkey to the south, and along Via Pontica, one of the bird migration routes from Africa to Europe [36,37]. Antibodies were not detected by either method in the provinces of Varna and Dobrich, also located along the Black Sea coast in northeastern Bulgaria. Previous human serological studies by ELISA detected antibodies in 3.33% of sera tested in Varna Province and 2.47% in Burgas Province [24], as well as antibodies in some migratory bird species in Burgas Province [38]. The negative results in our study are probably due to the relatively small number of tested serum samples from horses, 61 in total for the three provinces, which is a limitation of this study. The absence of neutralizing antibodies against WNV in the provinces of Stara Zagora and Smolyan was confirmed in our study of archival sera from donkeys and mules collected in 2015 [22].

The percentage of WNV seroprevalence was highest in the Northwestern region of Bulgaria, 22% on average (Table 2), ranging from 0% in Vratsa Province to 28.21% in Montana Province (Valchedrum Municipality). Moreover, the seroprevalence ranged from 18.2% to 50.0%, with neutralizing antibody titers from 1:10 to 1:80 in individual localities of the municipality. The municipality of Valchedrum borders the Danube River to the north, offering optimal conditions for the development of WNV vectors and favoring the transmission cycle, as well as Romania, where a large number of outbreaks and human deaths have been reported in the last 20 years [39]. However, we should further expand our study on samples collected from other regions bordering the Danube River to confirm this finding. A brief report on WNV seroprevalence in samples from 48 horses in September–November 2022 in northern Bulgaria showed 23% seropositivity [40]. However, it is not possible to compare and discuss these results against ours, since the study [40] reported only testing via ELISA and lacked sufficient information about the location and status of the tested animals.

We also found neutralizing antibodies in the Southwestern and South Central regions of Bulgaria, 2.1% and 1.2%, respectively. It is not surprising to find positive findings in our study in the Southwestern (Sofia city, Sofia Province, Blagoevgrad) and South Central (Haskovo Province) regions of Bulgaria. These areas also lie along the migration routes of wild birds from Africa through our country to other regions of Europe, Via Aristotelis and Via Balkanica, respectively [36]. Blagoevgrad Province borders Greece to the south, and the settlement in Haskovo Province from where the samples were obtained is located in close proximity to the borders of Greece and Turkey to the south. A nationwide screening study of WNV seroprevalence in humans from all 28 provinces in Bulgaria [23] found the highest rates in Sofia Province, 10% by ELISA in 2015. Another ELISA study 3 years later [24] found 1.25% seroprevalence in Sofia city and Sofia Province, 1.67% in Blagoevgrad Province and 1.43% in Haskovo Province. These reports confirm that WNV circulates among humans in Bulgaria and is spreading to new areas.

A seroprevalence against WNV of 53.33% was found in horses in southeastern Romania, a northern neighbor of Bulgaria, ranging from 0% to 75% in the Buzău (40%) and Tulcea (75%) Counties [41]. Another study [42] reported a seroprevalence of 33.53% in horses tested again in southeastern Romania in areas along the lower delta of the Danube River. Of note, in both reports, VNT was not used to confirm the ELISA-positive cases. Surveys in Serbia, which borders Bulgaria to the northwest, have shown intense circulation of WNV in the last decade. A study of horse sera obtained in 2011–2013 [43] found seropositivity ranging from 26.3% by ‘in-house’ IgG ELISA to 20.9% by commercial ELISA. A 2018 study [44] tested 2511 horse serum samples and found 44 positives (1.75%) for anti-WNV IgM antibodies. WNV is also widespread in other countries neighboring Bulgaria, such as Turkey, including its European territory [45,46], Greece [47] and North Macedonia [48]. These data provide evidence that WNV circulates in the Balkan Peninsula and is likely spread via wild birds and mosquitoes that can cross national borders, suggesting an endemic infection in the region.

Our seroprevalence data are in agreement with a previous report of 5.3% for WNV and 0% for Usutu virus in eastern Austria [49]. The authors recorded 15.5% for TBEV, which was not investigated in our study. TBEV infection has also been detected in other studies in Europe [50,51]. A human serological study in all 28 provinces in our country demonstrated a seroprevalence of 0.6% in 6 provinces by an ELISA kit [23]. Therefore, we could hypothesize that serum samples positive by ELISA alone could contain antibodies against this virus. In support of this hypothesis, ixodid ticks, which are vectors of TBEV, are widespread in our geographical area [50], and there have been sporadic human cases in Bulgaria [52]. The WNV seroprevalence data in our study are also similar to the 5.8% average recorded in horses in eastern Germany, ranging from 0% in Dresden to 14% in Elbe-Elster [19]. In contrast, Usutu virus (7.5%) circulation was found there.

IgM antibodies were not detected in the tested serum samples. Since IgM antibodies are detectable up to about the third month of encounter with the virus [53], their absence in positive sera indicates a past infection. The IgG neutralization titers, ranging from 1:10 to 1:80, in unvaccinated horses suggest the circulation of WNV in the tested provinces and regions of the country. Because of the low antibody titers (only four animals with 1:40 to 1:80), we could assume that infection in most animals occurred during the last season, since IgG antibodies circulate for about 1.3 years before their titers decline [54].

No attempts were made to detect WNV nucleic acids (NAs) in the collected samples in this study. There is a minimum likelihood to isolate nucleic acids in serum samples from equids, as confirmed in previous studies, where no NAs were isolated in any of the 348 samples in Austria [49] or the 437 samples in Germany [20].

Another objective was to analyze some risk factors that might influence WNV seropositivity. At the individual level, we found no risk associated with the age (*p* = 0.9125), species (horses, donkeys) (*p* = 0.9778) or sex of the animals tested (*p* = 0.6417). Seropositivity was not affected by age in other studies [18,19,55,56,57]. In contrast, Selim et al. [58] found a higher risk in horses aged ≥15 years, and Hassine et al. [59] found that increasing age by 5 years resulted in changes in seropositivity odds by a factor of 1.32. The type of animal and their sex were not relevant to risk in other studies [59], and no differentiation between types of equids was made [19]. Logistic regression analysis identified the region, elevation of the location, type of housing and breed as risk factors for WNV. As discussed above, the region is an important factor for seropositivity because certain environmental conditions favor vector growth and transmission cycle closure. The highest location with a positive sample in this study was at an altitude of 1100 m, while the majority of positive samples originated from lowland locations, with a risk threshold of ≤109 m. Our result is in agreement with the data from a previous report [60], which observed that an altitude of over 1700 m has a negative impact on the *Culex pipiens* population. Thoroughbreds were found to have a lower risk of WNV infection, which is directly related to the way they are kept, i.e., mainly in stables, where vector access is difficult, with shorter stays of the animals outdoors compared to those kept outdoors with shelter. This finding is similar to the risk of seropositivity analyzed for horses in East Germany [19].

This study has some limitations. It was not possible to obtain samples from all provinces in Bulgaria, or an equivalent number of samples per region. The lack of animal passports and the collection of data from elderly owners or keepers may have introduced some deviations from the actual age of the tested equids. The set of seropositivity risk factors that were analyzed was incomplete.

## 5. Conclusions

In this study, we detected virus-neutralizing antibodies in serum samples from equids that had had a permanent habitat over the last 2 years, allowing us to conclude that equids in Bulgaria are at risk of WNV in the studied provinces and regions, and that the infection is autochthonous. The serum samples that had an inhibition percentage over 90.76 but a negative result for WNV and Usutu virus-neutralizing antibodies suggest that there are probably other genetically similar orthoflaviviruses in circulation. This needs to be explored further in future studies. The present work fills a gap in serological testing for WNV and Usutu virus in sentinel hosts in Bulgaria (mainly horses), which can be used as an indicator to assess the risk in humans in the relevant provinces and regions.

## Figures and Tables

**Figure 1 vetsci-11-00209-f001:**
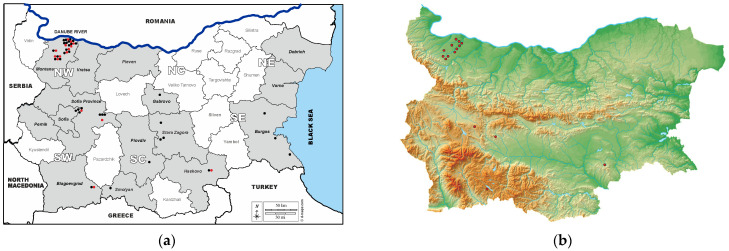
(**a**). Map of Bulgaria showing sampling provinces and ELISA- and VNT-positive samples within provinces. Note: NW—Northwestern; NC—North Central; NE—Northeastern; SW—Southwestern; SC—South Central, SE—Southeastern. Sampling provinces are shown in gray; the provinces from which samples were not collected are white. Black circles indicate the number of WNV-ELISA-seropositive samples and red circles indicate the number of WNV-VNT-seropositive samples in each province. (**b**). Elevation map of Bulgaria showing the distribution of WNV-VNT-seropositive samples.

**Figure 2 vetsci-11-00209-f002:**
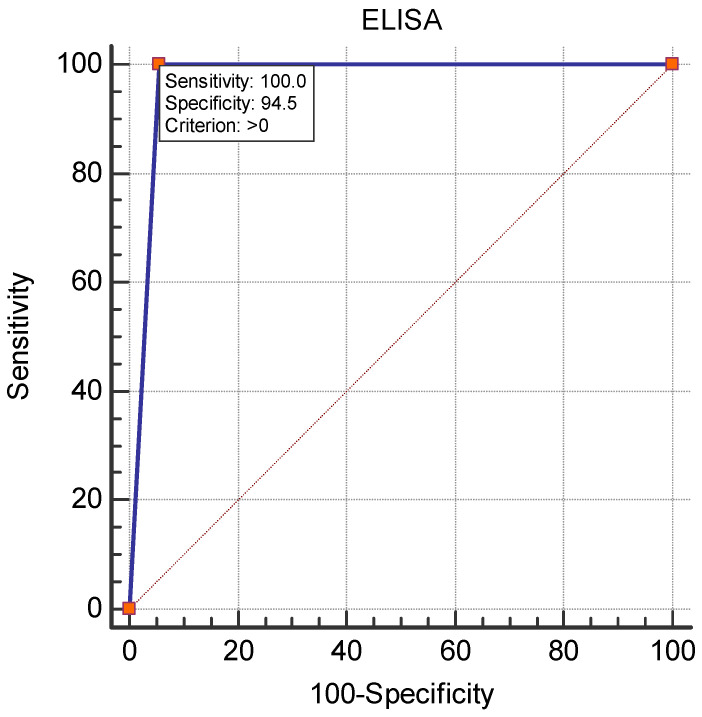
ROC curve analysis for the diagnostic performance of c-ELISA.

**Figure 3 vetsci-11-00209-f003:**
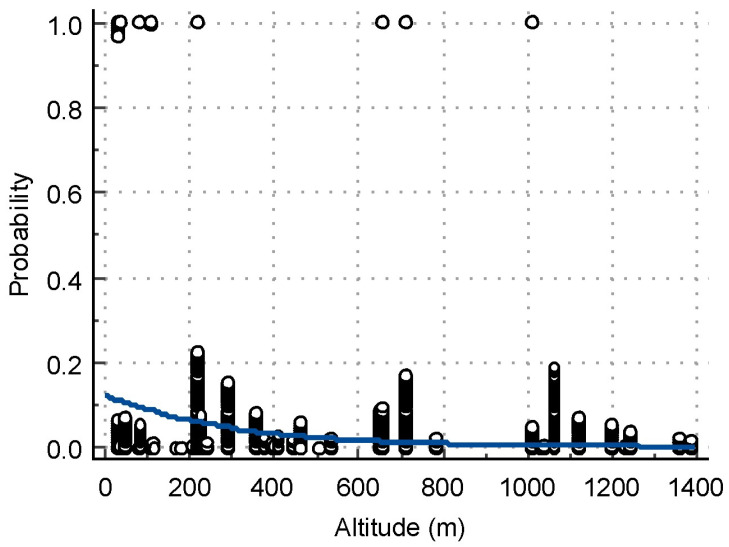
Logistic regression plot of the relationship between the altitude of sampling sites as an independent variable (*X*-axis) and the probability of having a positive VN test (*Y*-axis).

**Table 1 vetsci-11-00209-t001:** Seroprevalence of WNV in the studied provinces in Bulgaria.

Province	cELISA	VNT-WNV
Stara Zagora	3/34 (8.82% [1.82–25.79])	0/34 (0%)
Burgas	3/40 (7.5% [1.55–21.92])	0/40 (0%)
Haskovo	1/66 (1.52% [0.04–8.44])	1/66 (1.52% [0.04–8.44])
Plovdiv	1/1 (100%)	0/1 (0%)
Smolyan	1/18 (5.56% [0.14–30.95])	0/18 (0%)
Blagoevgrad	1/39 (2.56% [0.07–14.29])	1/39 (2.56% [0.07–14.29])
Pernik	0/5 (0%)	0/5 (0%)
Sofia city	4/26 (15.38% [4.19–39.39])	1/26 (3.85% [0.097–21.43])
Sofia Province	3/74 (4.05% [0.84–11.85])	1/74 (1.35% [0.03–7.53])
Montana	17/39 (43.59% [25.39–69.79])	11/39 (28.21% [14.08–50.47])
Vratsa	0/11 (0%)	0/11 (0%)
Pleven	0/1 (0%)	0/1 (0%)
Gabrovo	1/3 (33.33% [0.84–185])	0/3 (0%)
Dobrich	0/11 (0%)	0/11 (0%)
Varna	0/10 (0%)	0/10 (0%)
**Total**	**35/378 (9.26% [6.45–12.88])**	**15/378 (3.97% [**2.22–6.55**])**

95% confidence intervals are reported within squared brackets. cELISA, competitive enzyme-linked immunosorbent assay; WNV, West Nile virus; VNT, virus neutralization test.

**Table 2 vetsci-11-00209-t002:** VNT-WNV seroprevalence in 378 serum samples from horses and donkeys in Bulgaria.

Parameter		WNV Neg (n = 363) Median (IQR) or Percentage (%)	WNV Pos (n = 15) Median (IQR) or Percentage (%)	*p*-Value
**Region**	Northwestern	39 (78.0)	11 (22.0)	<0.0001
	North Central	4 (100.0)	−	
	Northeastern	21 (100.0)	−	
	Southwestern	141 (97.9)	3 (2.1)	
	South Central	84 (98.8)	1 (1.2)	
	Southeastern	74 (100.0)	−	
**Age, years**		9 (5–14)	10 (7.25–10)	0.9125
**Altitude, m**		407 (217–764.75)	77 (31–190)	<0.0001
**Species**	donkeys	62 (96.9)	2 (3.1)	0.9778
	horses	301 (95.9)	13 (4.1)	
**Breed**	mixed	203 (94.0)	13 (6.0)	0.0365
	purebred	160 (98.8)	2 (1.2)	
**Sex**	female	231 (95.5)	11 (4.5)	0.6417
	male	130 (97.0)	4 (3.0)	
**Type of housing**	outdoor	126 (95.5)	2 (4.5)	<0.0001
	shelter	15 (75.0)	5 (25.0)	
	indoor (stable)	222 (96.5)	8 (3.5)	
**Month**	July	58 (98.3)	1 (1.7)	0.0150
	August	163 (92.6)	13 (7.4)	
	September	58 (98.3)	1 (1.7)	
	October	84 (100.0)	−	

**Table 3 vetsci-11-00209-t003:** Statistical analysis of risk factors for WNV infection among equids in Bulgaria.

Variable		Odds Ratio (OR)	95% CI of OR	*p*-Value
**Region**	Southwestern	−		
	South Central	0.5595	0.0573 to 5.4663	0.6175
	Southeastern	0.0000		0.9940
	**Northwestern**	13.2564	3.5237 to 49.8711	0.0001
	North Central	0.0000		0.9986
	Northeastern	0.0000		0.9972
**Altitude**	**≤109 m**	**16.4474**	**5.0454 to 53.4759**	**<0.0001**
	>109 m	−		
**Breed**	mixed	−		
	**purebred**	**0.1952**	**0.0434 to 0.8775**	0.0331
**Type of housing**	outdoor	−		
	**shelter**	**21.0000**	**3.7415 to 117.8660**	**0.0005**
	indoor (stable)	2.2703	0.4748 to 10.8565	0.3045

## Data Availability

Data related to this study are available within the article.

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
