# Peer review of "Seroprevalence of West Nile Virus among Equids in Bulgaria in 2022 and Assessment of Some Risk Factors"

_vetsci, 2024, doi:10.3390/vetsci11050209_

Round 1
Reviewer 1 Report
Comments and Suggestions for Authors
Thank you for the opportunity to review the manuscript entitled “Seroprevalence of West Nile virus among equids in Bulgaria in 2022 and assessment of some risk factors” by Rusenova et al.
The manuscript describes the prevalence of antibodies against WNV in equids in Bulgaria, The manuscript is well written and expands on current understanding of WNV distribution in Bulgaria. The manuscript is aware of its limitations and discussed the sample set distribution as a limitation. Could the authors possibly comment on the distribution of Equidae in Bulgaria to possibly highlight any district variations.
A few comments to the authors:
Line 52, Culex pipiens L. complex refers to recombinations of the species however all species can carry and transmit WNV, please clarify why L. complex specifically cited.
Line 78, please remove etc. as citations provided of examples.
Line 105, please explain further the exclusion criteria as it is not fully clear if excluded movements to or from regions.
Line 116 and throughout, the use of ‘doubtful’ should be replaced with ‘inconclusive’ which is a more commonly used term for results that occur in the ‘grey zone’ if assays.
Line 132, confirm how CPE observed, I assume microscopically?
Line 135, please define cELISA
Line 138, please define ROC, it is defined at line 146 but should be defined at the first use
Table 1, please add the districts that were negative to the table
Figure 1, Please grey out districts from which samples were not received, it was difficult to interpret which were the negative districts or which were not tested without referring to Table 1 or supplementary information. It may also aid visualisation of results if the altitude of Bulgaria could also be overlaid in Figure 1.
Line 191, a correlate between altitude and WNV was established, could this data be represented graphically as a correlation curve, was it linear?
Line 223, not all samples were sent for VNT therefore can not draw conclusions about distribution of Usutu from this testing.
Line 245, please change ones to regions
Lines 264-267, this conclusion can not be drawn as only one region bordering Romania had positive samples, the other region bordering had one sample. Amend to note limitation of this conclusion in that only one region bordering Romania was positive.
Comments on the Quality of English LanguageGood English.
Author Response
Please, see the attachment.

Reviewer 2 Report
Comments and Suggestions for Authors
The manuscript entitled “Seroprevalence of West Nile virus among equids in Bulgaria in 2022 and assessment of some risk factors” by Rusenova et al., analyses the seroprevalences of WNV in 15 districts of Bulgaria. The study also examines the possible presence of USUV in those areas and the different risk factors associated with WNV infections in equids. In addition the manuscript is clearly written, with an informative introduction and a valuable discussion, and data are clearly presented. However, methodology regarding VNT must be re-examined and in consequence, the rest of the following analyses.
Minor comments for improvement:
-Line 47: please replace genus Flavivirur for Orthoflavivirus, as in 2023 genus and species nomenclature within the virus family Flaviviridae changed https://pubmed.ncbi.nlm.nih.gov/37561168/. Please also include an updated reference.
- Line 54: I would suggest more recent references, as they are from a decade ago.
-Fig.1: 3 Blue circles (ELISA) and 3 red circles (VNT) in Sofia province seem that they should be in Sofia, as they belong to Sofia district. It seems that these positive samples were taken in the border between Sofia province and Sofia. Could you please include these samples in Sofia district?
- Line 281: correct Bulgaira.
- Line 284: correct amont
- I recommend reducing the number of references, as 64 it is extremely high.
- Some of the references are outdated. I strongly recommend using more up-dated ones.
- Did you obtain any doubtful results by the initial competitive ELISA method? How did you proceed in those cases? Please indicate in the text
Major concerns:
- M&M: indicate the interpretation of VNT results when performing WNV and USUV neutralization assays, as it was done in the ELISA assays. Also indicate when a sample is positive for WNV or USUV or when it indicates the presence of an undetermined flavivirus.
Did authors obtained doubtful samples by ELISA methods? Why did authors not include doubtful ELISA samples in the VNT assays? As USUV is circulating in the area, there is a possibility that doubtful samples could be due to USUV presence.
According to WOAH Manual (also cited by authors), a titer of 1:5 is considered as negative sample “A VN titre greater or equal 1/10 is usually considered specific for WNV”. Following this WOAH indications, only 15 of the tested samples would be positive instead of 21. Thus, WNV seroprevalence in the region is estimated to be 3.98%.
On the other hand, the following WOAH indication must also be taken into account: “Cross-reactivity also exists between WNV and the Usutu virus however, specific antibodies can be attributed to one or the other virus as there is a neutralising titre of fourfold or higher for one virus compared with the other when independently tested”. Therefore, in this study it is also necessary to know the results of the USUV VNT, with the aim of correctly interpreting the general VNT results.
Both facts implicate to remake all the analysis and the results presented in this manuscript, or, at least, indicating in a clear way in the text, so readers can understand the procedures.
Author Response
Please, see the attachment.

Round 2
Reviewer 2 Report
Comments and Suggestions for Authors
Authors have improved the manuscript in some parts as requested. Moreover, the methodology and the way of analyzing the data are well considered. However, there are still some inconsistencies that they should correct prior publishing, particularly those related to the VNT technique and the following results and conclusions that they obtain.
Authors cited “Di Gennaro et al., 2014 (https://doi.org/10.1128/CVI.00426-14)“ as VNT-procedure reference. In this article, it is indicated that VNT starts in 1:5 of titer, but positive samples are those with a minimum titer of 1:10. “The serum samples were inactivated at 56°C for 30 min. Starting from a dilution titer of 1:5, serial 2-fold dilutions were made in microtiter plates, and 100 tissue culture infective doses (TCID) of antigen were added to each dilution […], and the antibody titer was defined as the reciprocal of the highest dilution of the serum that showed 100% neutralization. Positive and negative control sera were included in each plate. Sera with a titer of 1:10 were considered positive”.
In fact, it is usual to start with 1:5 with the aim of confirming USUV negative samples. For instance, if you obtain 1:10 WNV positive sera, you need to confirm that 1:5 USUV is also negative, otherwise you are not able to confirm that this sample is WNV positive.
In this article, whether authors want to discriminate between WNV or USUV positive sera, they should start in 1:2.5 titer, because a four-fold higher titer is required to provide a positive interpretation, as they comment in their response: “the sample was interpreted as positive for the virus with a four-fold higher titer compared to the other tested virus”.
For that reasons, I recommend authors to consider 1:10 as the starting positive dilution, changing the WNV seroprevalence to 3.98%, accordingly, they should modify the subsequent results. On the other hand, if they want to consider 1:5 as positive, they should perform again the VNT assay in those 1:5 positive samples, by testing 1:2.5 dilution of WNV and USUV with the aim of discriminating any possible cross-reaction.
Author Response
Dear reviewer,
Thank you once again for your valuable comments. We followed your suggestion and recalculated the results, to obtain 3.97% (3.968) seropositivity. We hope you find this version correct and suitable for publication.
Kind regards,
The corresponding author
The corresponding author
Round 3
Reviewer 2 Report
Comments and Suggestions for Authors.
Author Response
Dear reviewer,
Thank you very much for the positive evaluation of our revised version.
Kind regards